# An Open-Source Pipeline for Processing Direct Infusion Mass Spectrometry Data of the Human Plasma Metabolome

**DOI:** 10.3390/metabo12080768

**Published:** 2022-08-21

**Authors:** Anna Kozlova, Timur Shkrigunov, Semyon Gusev, Maria Guseva, Elena Ponomarenko, Andrey Lisitsa

**Affiliations:** 1Institute of Biomedical Chemistry, 119121 Moscow, Russia; 2Center of Life Sciences, Skolkovo Institute of Science and Technology (Skoltech), 121205 Skolkovo, Russia

**Keywords:** metabolomics, direct infusion mass-spectrometry, MALDIquant, MetaboAnalyst, MetaboAnalystR, Mummichog

## Abstract

Direct infusion mass spectrometry (DIMS) is growing in popularity as an effective method for the screening of biological samples in clinical metabolomics. Being quick to execute, DIMS generally requires special skills when interpreting the results of measurements. By inspecting the similarities between two-dimensional electrospray ionization with quadrupole time-of-flight (ESI-QTOF) and matrix-assisted laser desorption/ionization (MALDI) mass spectra, the pipeline for processing QTOF mass spectra using open-source packages (MALDIquant, MSnbase and MetaboAnalystR) was tested. Previously, all algorithmic workflows have relied on the application of software either provided by a vendor or privately developed by enthusiasts. Here, we computationally examined two ways of interpreting the DIMS results of human blood metabolomic profiling. The studied spectra were acquired using ESI-QTOF maXis Impact II (Bruker Daltonics, Billerica, MA, USA), then pre-processed using COMPASS/DataAnalysis commercial software and mapped onto the metabolites using in-lab-developed MatLab scripts. Alternatively, in this work we used the open-source packages MALDIquant, for spectrum pre-processing, and MetaboAnalystR, for data interpretation, instead of the low-availability commercial and home-made tools. Using a set of 100 plasma samples (20 from volunteers with normal body mass index and 80 from patients at different stages of obesity), we observed a high degree of concordance in annotated metabolic pathways between the proprietary DataAnalysis/MatLab pipeline and our freely available solution.

## 1. Introduction

Direct infusion mass spectrometry (DIMS) has become the method of choice for characterizing human metabolomes in just a few minutes. Comparatively, liquid or gas chromatography mass spectrometry (LC-MS/GC-MS) typically require 15 to 30 min for analysis. The time consumption imposes a significant limitation in screening the risks of diseases. To enhance the capacity of mass spectrometry, the DIMS method was proposed as an alternative to GC or LC-MS. Here, samples are directly sent to the mass detector without prior LC separation [1,2,3].

The limitation of the DIMS method is related to the interpretation of the resulting mass spectra. It is often extremely difficult to reliably identify which metabolites are present in the biological sample under study. According to the Human Metabolome Database (HMDB [4]), about 3000 endogenous metabolites with the status of “Detected and Quantified” or “Detected but not Quantified” have been detected in human blood via mass spectrometry using electrospray ionization. More than 20,000 metabolites in HMDB carry the status of “Expected but not Quantified” or “Predicted”. That is just a small number of all the compounds involved in cellular metabolism that have been experimentally found in human plasma.

In order to increase the number of metabolites identified in plasma, a pipeline for processing direct infusion mass-spectrometric data—DataAnalysis/MatLab—was proposed by Lokhov and colleagues [5]. The pipeline involves obtaining lists of masses in the COMPASS DataAnalysis program and comparing the masses found in a series of samples with the author’s alignment algorithm implemented in MatLab [6]. The algorithm created in 2011 involved building a matrix of mass-spectrometric peaks, then calculating the correlation coefficient for each row of the matrix and combining rows with a negative correlation. After alignment, the peaks were annotated, taking into account the biological context. The annotation algorithm was implemented in MatLab by Rogers and coauthors [7], and later improved [8].

The DataAnalysis/MatLab approach [5] for processing metabolomic direct infusion mass spectra is primarily based on the capabilities of the commercial program COMPASS DataAnalysis (Bruker Daltonics, Billerica, MA, USA). Usually, the COMPASS program allows the detection of peaks that can correspond to low-molecular-weight compounds. The peak annotation of mass spectra has also been performed with the web tool MASSTrix [9]. However, for the subsequent statistical analysis of annotations, the MatLab development environment (MathWorks, MA, USA) is required. Some home-made scripts implemented in the MatLab environment are not available for installation through repositories. We collected some of the original DIMS data from open repositories, including DI-ESI-QTOF datasets in MetaboLights [10] or the Metabolomics Workbench [11]. These data show that preprocessing methods are limited, either by the application of commercial programs or due to the lack of laboratory scripts shared on GitHub. As a consequence, it is currently impossible to repeat the computational experiment for determining the metabolomic profile obtained by direct infusion mass spectrometry. This significantly distinguishes metabolomics from proteomics and genomics, for which examples of public software for data processing and analysis have been developed, such as MaxQuant [12], SearchGUI [13] and Guppy basecaller [14].

This work aims to develop an alternative approach for processing direct infusion mass spectra based on freely available software. The authors propose the MALDIquant/Mummichog pipeline based on the functionality of the publicly available MALDIquant package [15]. Our method, similar to that described above [5], implements the following steps (see Figure 1 and Appendix A). First, mass-spectrometric scans are combined into one consensus mass spectrum, and then intensity smoothing, baseline removal, peak detection and equalization are performed. A comparison of m/z values with metabolites and metabolic pathways is carried out using the MetaboAnalystR–Mummichog module.

In this work, we rely on the equivalence of the algorithms for processing the mass spectra obtained on MS platforms with two different types of ionization: matrix-activated and electrospray. The MALDIquant program, which is part of the freely distributed R language package, is applicable to matrix-activated ionization mass spectra. The mass spectra obtained by electrospray ionization (ESI) with direct infusion into the ion source of the QTOF instrument are similar to MALDI-MS spectra—in both cases, the sample is not subjected to chromatographic separation.

For a comparative analysis of the DataAnalysis/MatLab and MALDIquant/Mummichog approaches, previously obtained by Lokhov and colleagues, data pertaining to the metabolomic profiling of blood plasma have been used [5]. In that study the mass spectra were acquired on a high-resolution hybrid quadrupole time-of-flight mass analyzer (QTOF, electrospray ionization), maXis Impact II (Bruker Daltonics, Billerica, MA, USA).

## 2. Materials and Methods

A comparative analysis of the DataAnalysis/MatLab and MALDIquant/Mummichog approaches was performed by comparing the number of peaks, the number of metabolites, and the types of annotated metabolic pathways at various stages of data processing. A Venn diagram of the approaches is shown in Figure 1.

Figure 1 shows the scheme of processing the mass-spectrometric data obtained by the DIMS method. The same data (100 spectra) are processed in two ways: on the left (purple), the DataAnalysis/MatLab approach based on commercial software is applied, and on the right (green) the MALDIquant/Mummichog approach based on the use of free software is applied. The steps following the annotation of metabolites and biochemical pathways are the same both for the previously developed approach [5] and our approach (see Appendix A).

### 2.1. Initial Data

The description of 100 volunteers’ blood plasma metabolomic profiles is presented in an article by Lokhov and coauthors [5]. The Ethics Committee name and approval code are indicated in the text of the manuscript in section “2.1. Sample Collection” as a quote: “The study was approved by the relevant ethical review committee of the Federal Research Centre of Nutrition, Biotechnology and Food Safety (protocol #4 from 15 June 2018).” In that study, five groups with 20 samples in each (10 male and 10 female) were analyzed. The first group comprised healthy volunteers, the second group — volunteers with overweight, and the remaining three groups were categorized depending on the stage of obesity. After the extraction of the low-molecular-weight fraction from samples, the direct infusion was carried out and 60 scans were acquired, with one technical replicate in the mode of positively charged ions at an acquisition rate of 1 spectrum per second. As a result, we obtained 100 “*.d”-directories with a total volume of 5.12 GB (Figure 1). Each directory contained arrays with the values of mass-to-charge characteristics, the intensities of registered ions and information about the settings of the mass spectra acquisition mode.

### 2.2. Step 1. Conversion of “.d”-Directories Using MSConvert

For subsequent loading into MALDIquant, the initial mass-spectra (“.d”-directories) are transferred from the binary to the mzML format. Note that there is no such step in the case of using the COMPASS DataAnalysis program, since it allows one to immediately translate raw data into a text format of peak lists (see Figure 1, step 1). The mzML format, developed in 2009 as part of the Human Proteome Organization (HUPO) initiative, as a standard [16], contains much more information about the results of mass-spectrometric measurements than the peak list generated by the COMPASS DataAnalysis program.

The initial data were converted to the mzML format using the MSconvert program, which is one of the tools of the ProteoWizard software [17]. The conversion was performed in multi-threaded mode (Yandex Cloud, 8 vCPU, Intel Broadwell) with a cross-platform interface mono [18].

### 2.3. Step 2. Obtaining a Consensus Spectrum

After converting the mzML format data, the combineSpectra function from the MSnbase package [19] was used to combine several scans into a common consensus spectrum. The launching parameters of the combineSpectra and other addressed programs are given in Appendix A. The resulting merged spectra were written into new versions of the mzML files; from each file with several scans, one file with a consensus spectrum was obtained (Figure 1, step 2).

### 2.4. Step 3. Smoothing, Alignment and Peak Detection

To find the peaks in the consensus spectra, we smoothed the intensities using the MALDIquant::smoothIntensities function, specifying the method and the value of the smoothing window (see the parameters in Appendix A). The direct infusion mass spectra were aligned using the MALDIquant::alignSpectra function, then peak picking was performed with the MALDIquant::detectPeak function. When aligning mass spectra between samples, m/z values that differed by no more or equal to the tolerance value 0.001 were assigned to the same possible compound mass in daltons (Da). The intensities of the peaks that coincided within the m/z value mass tolerance were averaged using the MALDIquant::binPeaks method (options are indicated in Appendix A). The peak processing results were exported to plain text files for further annotation.

### 2.5. Step 4. Annotation of Peaks Based on Metabolic Pathways

The annotation was performed using the Mummichog module integrated into MetaboAnalystR. It was originally designed to annotate LC-MS m/z peaks based on the results of biochemical transformation prediction [20]. The use of Mummichog made it possible to take into account not only the metabolite, through the measured experimental value of m/z, but also an annotation of the most probable biotransformation products, including the products of typical enzymatic reactions and chemical adducts based on the isotopes of chemical elements. The results of the annotation were correlated with the library of biochemical processes in the KeGG knowledge base [21].

In the diagram (see Figure 1), peak annotation is indicated as a part of the general algorithm for the processing of direct infusion mass spectra. In the DataAnalysis/MatLab pipeline (the left part of Figure 1, highlighted in purple), this stage is labeled as “Annotation with biological context”. On the right part of Figure 1 (highlighted in green) within the MALDIquant/Mummichog pipeline, the corresponding stage is indicated as “MetaboAnalystR::Mummichog”.

Using Mummichog, we annotated the mass spectrometry peaks based on the biological context. As the input data, we used files obtained after exporting the results of the alignment and peak binning in step 3 (see Figure 1). When comparing the norm and stages of obesity groups, for each m/z value, the input file was supplemented with the column “*p*.value”. The *p*-values were calculated in the Statistical Analysis MetaboAnalyst 5.0 [22] program and used to characterize the significance of the peak intensity differences when two groups were compared. To calculate the *p*-value, a single-factor non-parametric Wilcoxon test was used.

Mummichog was run for the entire dataset obtained after processing 100 samples with the coarsened statistical significance parameter of intergroup comparison: *p*-value = 0.99.

### 2.6. Mapping of Metabolites to Biochemical Pathways

The step of mapping metabolites to biochemical pathways is shown in Appendix A. After the putative annotation of the m/z values set, the transition to metabolic pathways was carried out using the MetaboAnalyst 5.0 program [22]. Pathway information was taken from the KeGG [21] and SMPDB [23] databases. To compare the identified metabolites with pathways, the authors of this study, following the procedure of the group of Prof. Lokhov, used the enrichment analysis module MSEA (Enrichment Analysis, see options in Appendix A). The set of parameters is also equivalent to that used earlier [5].

### 2.7. Virtual Machine Yandex Cloud

In order to ensure the availability and reproducibility of the developed pipeline for processing mass-spectrometric data, a cloud computing approach was applied to the Yandex Cloud platform [24].

A virtual machine was created on the Yandex Cloud platform [24] to process direct infusion mass spectra (OC Ubuntu 20.04 LTS, 8 vCPU, 32 GB RAM, System HDD 20 GB; a 30 GB HDD was connected to store the initial experimental data and process results).

RStudio-server version 1.4.1106 (RStudio, MA, USA) was installed on the virtual machine to allow the use of the RStudio development environment through a web browser for working with the MALDIquant and MetaboAnalystR packages. The R language packages MALDIquant version 1.21, MALDIquantForeign version 0.13, MSnbase version 2.22.0 and MetaboAnalystR version 3.2.0 were installed in the RStudio interface. The initial files were loaded using the WinSCP program, version 5.19.6.

## 3. Results and Discussion

We aimed to determine whether the MALDI spectra processing tool was applicable to the data obtained by direct infusion with the electrospray ion source on a hybrid quadrupole time-of-flight mass analyzer (DI-ESI-QTOF). A routinely used pipeline for processing mass-spectrometric data (peak detection, annotation of compounds, mapping to metabolic pathways) was implemented using open-source programs [15,17,19,20,22]. The most highly represented (by the number of annotated compounds) metabolic pathways were compared between two approaches: DataAnalysis/MatLab (commercial software) and MALDIquant/Mummichog (open-source solution).

When using the open-source solution, the essential criterion was comparability with the published approach [5], in terms of the detected number of peaks. Recently, the DEIMoS package appeared in the open-source domain [25]. This represents a significant step in metabolomics from proprietary data and algorithms to reproducible pipelines. We inspected DEIMoS using our dataset. Due to the wide functionality of the package, we failed to cope with baseline correction and peak assembly, which are simple in MALDIquant. Additionally, XCMS has gained well-deserved popularity and was originally developed for processing LC-MS data. We tested the pipeline in XCMS for processing the direct infusion mass spectrometry FT-ICR data [26]. Utilizing the XCMS package, we detected 447 ± 141 peaks per sample, which is substantially lower than when processing DIMS spectra in the DataAnalysis program (see Table 1). The MALDIquant package, which was originally developed for two-dimensional mass spectrometry data, led to the detection of 9274 ± 297 peaks and so, for further analysis, MALDIquant was chosen.

Table 1 presents the results of employing combinations of moving average and Savitzky–Golay filters with two noise-reduction methods: median absolute deviation (MAD) and SuperSmoother. Using the moving average method as an example, we have shown the dependences between the numbers of detected peaks and the smoothing half-window size parameter (hws). With coarsened smoothing (hws = 1), a relatively larger number of peaks was obtained—up to 10,000 on average. With a wider window (hws = 4), the number of peaks was 30–40% lower than the reference value. However, if we smoothed not with a moving average, but with the Savitzky–Golay algorithm, then the number of peaks and the range of the average value (9274 ± 297) almost exactly matched the data obtained using DataAnalysis (9333 peaks). We stopped at the Savitzky–Golay method in the application of the MALDIquant approach for further analysis (indicated in Table 1 through the context). Note that the coincident number of peaks does not mean that these peaks obtained by different tools result in identical m/z values. Therefore, in order to correctly compare the previously published approach [5] with the proposed pipeline, it was necessary to map the metabolomic profiles to biochemical pathways.

Figure 2a shows that the DataAnalysis/MatLab approach made it possible to annotate the chemical names for 390 metabolites (*p*-value < 0.01) in plasma samples from volunteers with the third stage of obesity. The developed MALDIquant/Mummichog algorithm made it possible to annotate 920 metabolites (with a coarsened *p*-value = 0.99) in the same data. When setting a cutoff *p*-value < 0.01 (Wilcoxon test result when comparing volunteers and patients) in our pipeline, we obtained only 7 metabolites (see below and Appendix A), so when building the Venn diagram, the set of 920 metabolites in MALDIquant/Mummichog, annotated with a coarsened *p*-value, was used (Figure 2a).

To compare the results of the approaches, the metabolites involved in the biosynthesis of steroid hormones and found in samples from patients in the third stage of obesity were selected as an example (see Figure 2b). The intersection of the sets of metabolites obtained by DataAnalysis/MatLab (*n* = 40) and MALDIquant/Mummichog (*n* = 59) equaled 50% (*n* = 33) of the total number of metabolites found by the two approaches (*n* = 66). Importantly, all nodal metabolites, i.e., the metabolites with the most edges, were successfully annotated using both approaches (Figure 2c,d).

One of the possible reasons for the incomplete intersection in the lists of metabolites is the errors made in determining the MALDIquant peaks. When using MALDI, it is common to collect signals of 10,000 laser shots to compute a consensus spectrum. Unlike MALDI, DIMS records only 60 spectra per minute, and this can affect the quality of peak recognition when building a consensus spectrum.

Another possible reason for the discrepancies in the sets of metabolites observed in Figure 2a,b is the difference in approaches in terms of the m/z value and the compound identifier in KeGG; in Mummichog, the variation range of m/z values is expanded due to the combinations of adducts [20].

Using the Statistical Analysis module of the MetaboAnalyst 5.0 web platform, we conducted a comparative statistical analysis to search for differential metabolic markers of obesity. A comparison was carried out using the Wilcoxon test regarding the values of the intensities of metabolites in healthy volunteers compared with, in one case, all stages of obesity, and in the second case, only the third stage. Appendix A includes 96 m/z values that differ significantly in intensity for the two listed cases. The table shows that the number of annotated metabolites is negligible compared to what was published in the previous article [5]. A total of nine m/z values were mapped to KeGG IDs. Among the values that arose when comparing the norm and all stages of obesity, there were seven metabolites, including glycans; orotic, glutamic and arachidonic acids; and nicotinamide ribotide (see Appendix A). Comparing the norm with the third stage of obesity, only four metabolites with statistically significant intensity differences were identified.

As can be seen from Appendix A, several metabolites may give the same m/z values when searched in the KeGG database. For example, the Mummichog program compared six different metabolites with a statistically significant peak (*p*-value < 0.01) characterized by m/z = 148.059. Among them, 7,12-Dimethylbenz[a]anthracene 5,6-oxide most likely has an exogenous origin, entering the body with plant foods (see Appendix A). Considering the sampling conditions (in the morning with an empty stomach), we can assume that this is an example of a false positive result, since the presence of plant metabolites in the studied human blood samples is unusual.

The negligible number of metabolites identified by statistical analysis is unsurprising for two reasons. Firstly, the absolute value of mass spectrum intensities in different samples is incomparable due to the absence of external or internal calibration. Incidentally, in the previous work [5], such calibration was carried out. Secondly, it is necessary to take into account the key feature of Mummichog’s work [20]. This algorithm is not intended for metabolite annotation, i.e., the matching of putative analytes to KeGG identifiers is essentially a by-product. The main problem solved by Mummichog is the prediction of metabolic pathways. Therefore, our further study did not compare m/z values or lists of KeGG identifiers, but instead analyzed the enrichment of certain metabolic pathways.

The results of the enrichment assessment obtained using the MetaboAnalyst 4.0/5.0 web platform are shown in Figure 3. The ranked lists of metabolic pathways obtained by the MSEA tool [27] are shown for two compared approaches: DataAnalysis/MatLab (Figure 3a) and MALDIquant/Mummichog (Figure 3b). Figure 3a shows the MSEA result obtained using the COMPASS DataAnalysis approach and the annotation algorithm in MatLab [6]. Figure 3b shows a diagram constructed in a similar way for a case in which MALDIquant was used in combination with Mummichog instead of DataAnalysis and MatLab.

The application of the DataAnalysis/MatLab approach made it possible to detect metabolic pathways with high-significance values (*p*-value < 0.01). The first of the metabolic pathways, steroid biosynthesis, differs in intensity between obese and normal cohorts by more than four orders of magnitude (see the Fold Enrichment parameter). The MALDIquant/Mummichog pipeline, however, did not allow for the identification of enriched pathways associated with obesity: statistical differences in the norm compared with patients were not significant (*p*-value > 0.2). For example, in Figure 3a, the metabolic pathway of steroid biosynthesis shows the highest confidence. However, Figure 3b shows that when using our approach to process DIMS data, this path has no statistical significance and barely reaches Fold Enrichment = 0.6.

In addition to MSEA, the MetaboAnalyst platform provides another tool for characterizing metabolomes. Figure 4 shows the results of the Pathway Analysis module, obtained by analyzing the results of processing the DIMS spectra in the frame of our MALDIquant/Mummichog pipeline.

The Y-axis in Figure 4a,b characterizes the relationship reliability of a compound group detected by mass spectrometers with one or another metabolic pathway. Two situations are depicted: people with a normal body mass index (Figure 4a) and patients suffering from stage III obesity (Figure 4b). The dots on the scatter plot correspond to pathways in which metabolites are found relatively more frequently than would be expected. It can be seen that the compounds measured using DIMS are more likely to be related to the primary bile acid metabolism.

Figure 4 indicates that the proposed data analysis solution does not reveal significant differences in the profiles of metabolic processes between the normal and obese cohorts; in both studied cases (Figure 4a,b), first place is held by the formation of primary bile acids, the second is the biosynthesis of steroids, the third is the biosynthesis of unsaturated fatty acids and then the metabolism of retinol and steroid hormones follows.

Although our method did not reveal the features of metabolic pathways characteristic of obesity, it is effective for exploring the metabolic pathways in normal and obese patients. This can be seen from an individual-level metabolic pathway comparison (Figure 5). Graphically, this comparison is shown in Figure 5, where the published results of MSEA are matched to the results of the Pathway Analysis. It is important to recall that a list of metabolites annotated by the Mummichog program was loaded into Pathway Analysis, provided that each m/z value was given a *p*-value = 0.99. The results of the Pathway Analysis are ranked in descending order of *p*-value, which in this case characterizes the probability of the misidentification of the metabolic pathway (see Figure 5b).

Figure 5a,b shows that, for example, bile acid biosynthesis is not unrelated to metabolic disorders in obesity (*p*-value < 10^−8^). If we compare the first five metabolic pathways shown in Figure 5b with the results of MSEA (see Figure 5a), taking into account the peculiarities of the biochemical origin of metabolites in the body, then steroid biosynthesis (“steroidogenesis”) can be attributed to “primary bile acid biosynthesis” and “steroid biosynthesis”, since the compounds included in both metabolic pathways are included by their chemical nature in the class of steroids and are derivatives of cholesterol.

Similarly, using the structural formulas of metabolites, the other two points from the set enrichment of metabolites (MSEA) were correlated. Using the MALDIquant/Mummichog method, the compounds involved in the metabolism of androgens, estrogens, androstenedione and estrone were identified. The listed compounds are steroid hormones and can be related to the biosynthesis of steroid hormones (Figure 5a,b). The statistical significance of the pathways determined using the Pathway Analysis module of the MetaboAnalyst 5.0 web platform is characterized by a *p*-value < 0.003. Such pathways include 16 to 60 metabolites, as shown in the “Match Status” column in Figure 5b. Interestingly, 16 compounds fell into the pathway labeled “Retinol metabolism” in the KeGG database. Retinol metabolism includes biochemical processes associated with the conversion of vitamin A in the human body. A number of articles [28,29] indicate that retinoic acids affect the differentiation of adipocytes in both white and brown adipose tissue. Returning to Figure 2c,d, it can be seen that the MALDIquant/Mummichog approach covered 19 more metabolites of the steroid hormone biosynthesis metabolic pathway than the COMPASS/MatLab approach.

## 4. Conclusions

Direct infusion ESI-QTOF mass spectrometry has made it possible to obtain a “snapshot” of a person’s phenotype through the blood, via its metabolomic profile. The blood metabolome reflects the genetically determined features of metabolism and changes in the biochemical processes of organ systems, as well as the lifestyle and potential habits of the individual. We have demonstrated the DIMS–data processing pipeline, using the example of a problem related to the metabolomics of obesity.

One of the more recently published data processing algorithms used for direct electrospray infusion mass spectrometry in connection with the problem of obesity involved preprocessing mass spectra using the COMPASS DataAnalysis software developed by Bruker Daltonics (Billerica, MA, USA) [5]. Previously, the same group showed [5,6,30] that the data obtained by the DIMS method reflect the human molecular phenotype, so this approach can be used for the laboratory diagnostics of socially significant diseases.

The process of obtaining mass spectra by direct infusion is similar to obtaining the spectra of matrix-activated ionization (MALDI-TOF). When using the DIMS method, a set of mass spectra separated by certain time intervals (1 s) is generated, similar to the process by which MALDI performs a series of laser strikes on a target, which it also does with time fixation. Both with direct electrospray infusion and with matrix-activated ionization, the system for recording the mass-to-charge characteristics of ions is the same: time-of-flight (TOF). Despite the fact that the QTOF mass analyzer (maXis) operates in a tandem mode, when using DIMS, the second quadrupole (q2) is not used, and metabolite ions enter the TOF analyzer without fragmentation in exactly the same way as in experiments using MALDI-TOF. We have shown that the computer processing method developed for the MALDI proteomic platform can also be used in the analysis of the metabolome obtained by direct infusion.

To investigate the metabolome, we applied free MALDIquant processing programs to DIMS data and demonstrated the following:

(a) MALDIquant makes it possible to determine the comparable values of peaks in mass spectra (9333 peaks in COMPASS DataAnalysis compared to 9227 peaks in MALDIquant);

(b) the use of MALDIquant together with the Mummichog module of the MetaboAnalystR package provided > 50% coverage of metabolites from previously published results [5], in which metabolites were identified with commercial packages and home-made scripts;

(c) our approach, based on a public bank of programs, made it possible to determine the previously obtained metabolic pathways [5] associated with the biochemical features of obesity development.

However, our results differ significantly from those previously published [5]. When explaining the differences, it is important to note that we applied a non-standard method, using the same algorithm for processing mass spectra for laser and electrospray ionization. We used the Mummichog module as a peak annotation algorithm, which is designed for the direct transition from the mass-to-charge characteristics of ions to metabolic pathways. The problem of the unambiguous correct annotation of metabolites using the m/z value has not been solved. Our results also demonstrate the different degrees of reliability of the annotations made using the statistical approach (Mummichog) and the Bayesian approach [5]. Annotation based on biological context makes it possible to detect more accurate differences in metabolomic profiles compared to using the Mummichog statistical approach. However, these discrepancies are common in metabolite identification, and the two different approaches—Mummichog and the Bayesian approach—can be considered complementary.

With the appropriate normalization of the peak intensities, the proposed approach will allow the use of the DIMS data obtained with the QTOF instrument to determine differences associated with overweight and obesity. The approach has been implemented using a cloud service, which allows for replicating a virtual machine and reproducibly reanalyzing proprietary collections of DIMS data in order to build a human digital image [31].

## Figures and Tables

**Figure 1 metabolites-12-00768-f001:**
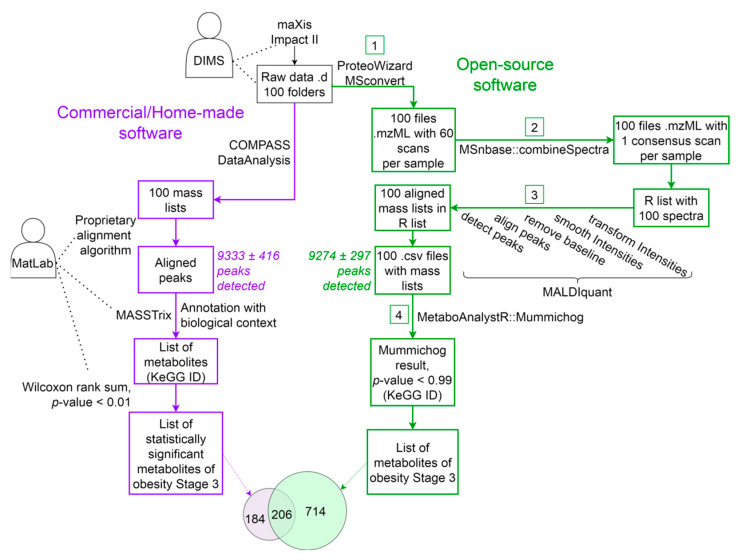
The order of launching programs for processing DIMS (ESI-QTOF, maXis Impact II, Bruker Daltonics) mass spectra. On the left (purple), a pipeline with the commercial COMPASS DataAnalysis program, and the MatLab development environment is shown. On the right (green)—the MALDIquant package together with MetaboAnalystR/Mummichog R language module. The numbers in the squares indicate the data processing steps described in the text.

**Figure 2 metabolites-12-00768-f002:**
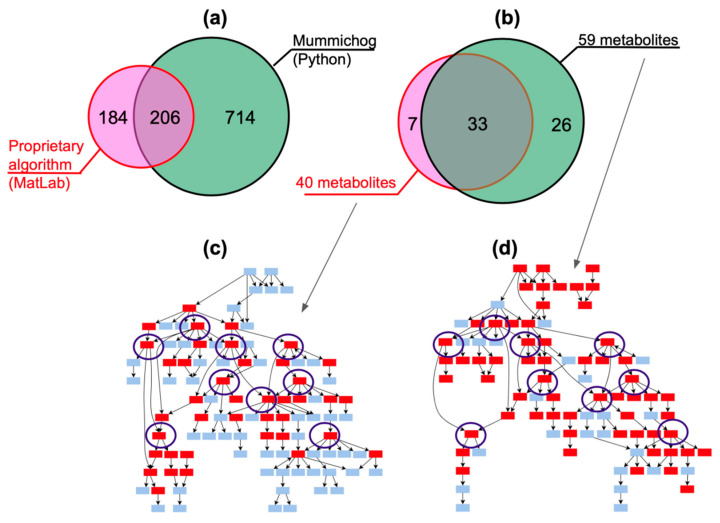
(**a**) KEGG-annotated metabolites related to the third stage of obesity (20 samples). The pink circle denotes the set of metabolites defined by the biological context, implemented in the MatLab development environment, followed by the application of the Wilcoxon test (*p*-value < 0.01). The green circle represents the set of metabolites annotated with the Mummichog algorithm (MetaboAnalystR package). (**b**) Metabolites related to the synthesis of steroid hormones. The pink circle is the number of metabolites found by the algorithm based on the biological context (MatLab script), the green circle is the number of metabolites found by Mummichog. (**c**,**d**) Steroidogenesis pathways with the indication of annotated (red boxes) and non-annotated (blue boxes) metabolites. Outlined boxes with KeGG IDs are identical nodal metabolites found using two compared approaches.

**Figure 3 metabolites-12-00768-f003:**
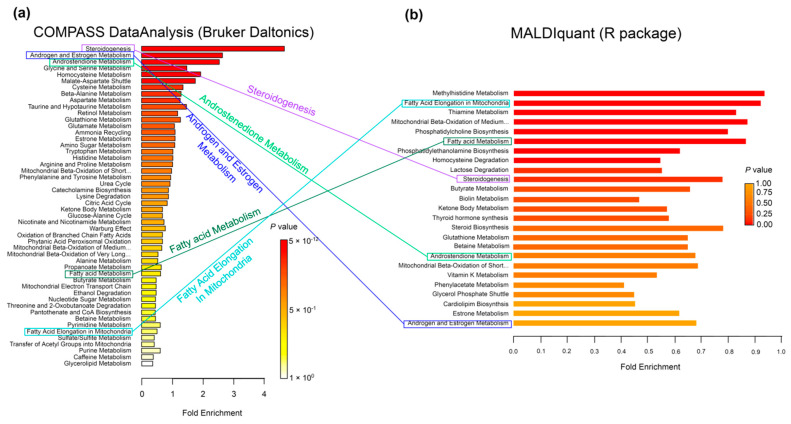
Metabolite set enrichment analysis (MSEA [27]): obesity metabolome, stage three. The results of the analysis of a set of metabolites in samples from patients with stage three obesity using (**a**) MetaboAnalyst web platform 4.0 [5] and (**b**) using MetaboAnalyst 5.0. The color gradient from red to yellow shows the level of significance (*p*-value), which characterizes the reliability of the detection of metabolic pathways based on DIMS data.

**Figure 4 metabolites-12-00768-f004:**
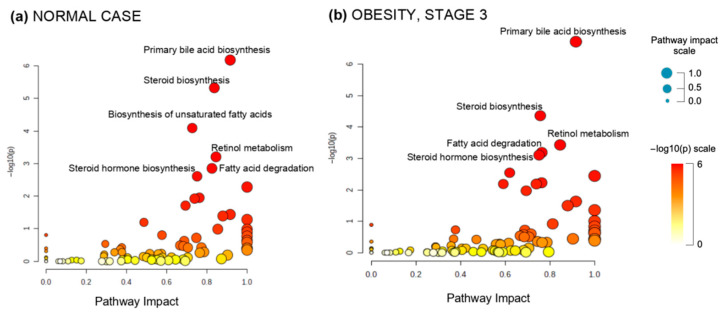
Metabolomic pathways profiles produced using Pathway Analysis module of MetaboAnalyst 5.0 web-platform: (**a**) normal—20 samples (**b**) third stage obesity—20 samples.

**Figure 5 metabolites-12-00768-f005:**
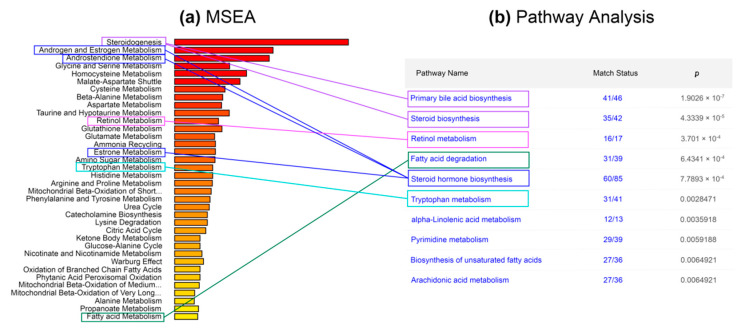
Enriched metabolic pathways in the third stage of obesity, MetaboAnalyst 4.0/5.0, web-version. (**a**) DIMS mass-spectra after processing with the DataAnalysis/MatLab pipeline were analyzed by the MSEA tool. (**b**) DIMS mass-spectra after MALDIquant/Mummichog pipeline processing were analyzed with the Pathway Analysis tool. “Match Status” indicates the number of annotated metabolites/number of known metabolites of the KEGG pathway.

**Table 1 metabolites-12-00768-t001:** The number of peaks detected using different intensity smoothing and noise estimation algorithms.

MALDIquant Algorithm		Number of Detected Peaks (Mean ± SD)		
Smoothing	Noise Estimation	MALDIquant	XCMS	COMPASS DataAnalysis
MovingAverage (hws = 2)	MAD ^1^	9908 ± 354	447 ± 141	9333 ± 416
	SuperSmoother ^2^	10768 ± 698		
MovingAverage (hws = 3)	MAD	8887 ± 144		
	SuperSmoother	8113 ± 261		
MovingAverage (hws = 4)	MAD	5778 ± 164		
	SuperSmoother	6728 ± 348		
SavitzkyGolay (hws = 4)	MAD	9274 ± 297		
	SuperSmoother	9834 ± 584		

^1^ MAD—median absolute deviation. ^2^ SuperSmoother—Friedman’s nonparametric regression estimator.

## Data Availability

The source code is available on https://github.com/Ministreliya131/An-OpenSource-Pipeline-for-Processing-Direct-Infusion-Mass-Spectrometry-Data (accessed on 20 August 2022). The data is available on MetaboLights public repository at www.ebi.ac.uk/metabolights/MTBLS5312 (accessed on 20 August 2022) with MTBLS5312 identifier and on Mendeley Data at http://dx.doi.org/10.17632/d2hrxnws6k.1 (accessed on 20 August).

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
