# Peer review of "An Open-Source Pipeline for Processing Direct Infusion Mass Spectrometry Data of the Human Plasma Metabolome"

_metabolites, 2022, doi:10.3390/metabo12080768_

Round 1
Reviewer 1 Report
The authors report the metabolomics analytical tool of mass spectra obtained using direct infusion mass spectrometry. The versatility of the method was well described and the developed tool seems to be useful. Generally, the manuscript is well prepared. However, I have a suggestion to add to the manuscript. In figure 2, the metabolites identified by previous and previous works were not matched completely. This discrepancy is common in metabolites identification and two different approaches can be considered as complementary.
One minor correction: ‘m/z’ was written as ‘m\z’ a few times.
Author Response
Reviewer: The authors report the metabolomics analytical tool of mass spectra obtained using direct infusion mass spectrometry. The versatility of the method was well described and the developed tool seems to be useful. Generally, the manuscript is well prepared.
Point 1: However, I have a suggestion to add to the manuscript. In figure 2, the metabolites identified by previous and previous works were not matched completely. This discrepancy is common in metabolites identification and two different approaches can be considered as complementary.
Response 1: Thank you for your significant remark. We included the following suggestion to the “Conclusions” section (Lines 442-444 ).
Point 2: One minor correction: ‘m/z’ was written as ‘m\z’ a few times.
Response 2: Sorry for this technical mistake. The whole text was rechecked.
Authors: We thank Reviewer for attention to our manuscript and positive feedback. The authors agree with the received comments and have taken them into account while preparing the new version of text.
Reviewer 2 Report
The authors present a pipeline of to process mass spectrometry data. The first comment is the benefit of direct infusion. It is not clear why to present such a software. It is also addressed there are several methods to process the data. The necessity of this work need clarified. The second is about the human blood plasma. The authors used the specific data for an illustration purpose. It still needs to present some results about the findings of the case metabolite data. Third is about the availability of the package will be important for this software. The authors need open the source code and deposit all the data and code in public repository. Thanks.
Author Response
Reviewer: The authors present a pipeline of to process mass spectrometry data.
Point 1: The first comment is the benefit of direct infusion. It is not clear why to present such a software. It is also addressed there are several methods to process the data. The necessity of this work need clarified.
Response 1: Deep thanks for your question. The question highlights two warnings: why do we need the direct infusion mass-spectrometry? If direct MS is not needed, therefore why extra bioinformatics methods (as we have developed) are needed to process presumably unnecessery data? The only advantage of DIMS is that is rapid, knowing that any rapidness is full of errors. In our article we gave for the first time an instrument to asses the level of errors for DIMS, that was our goal. Explicitly sharing your opinion at the same time we kindly ask to take off your comment that will recrute critical followeres to test the honesty of the DIMS.
Point 2: The second is about the human blood plasma. The authors used the specific data for an illustration purpose. It still needs to present some results about the findings of the case metabolite data.
Response 2: Thank you for valuable comment. The main purpose of our work was to develop and validate the open-source pipeline for processing the direct infusion mass-spectrometry data. The exisiting widely used non freely available solution was used for comparison (COMPASS DataAnalysis, MatLab). In the case metabolome data, both solutions have shown comparable results at the metabolic pathway level. We concluded that two different approaches can be considered as complementary. To stated more clearly we added to the “Conclusions” section (Lines 436-438) the following phrase: ”However, these discrepancies are common in metabolites identification and two different approaches - Mummichog and the Bayesian approach - can be considered as complementary”.
Point 3: Third is about the availability of the package will be important for this software. The authors need open the source code and deposit all the data and code in public repository.
Response 3: The authors concur with the Reviewer and support open access. The source code is now available on https://github.com/Ministreliya131/Direct-Infusion-Mass-Spectrometry-Data-Processing-Pipeline-with-MALDIquant-and-MetaboAnalystR. The data were deposited to the MetaboLights public repository on www.ebi.ac.uk/metabolights/MTBLS5312.
Authors: We thank the Reviewer for important remarks and comments and took them into account while preparing the revised version of manuscript.
Reviewer 3 Report
The authors are trying to develop an R-based method (MetaboAnalystR) for metabolome data processing. Compared with the existing related software, the software shows advantages in the data processing in the manuscript. The manuscript describes a possible open source metabolomics data analysis method, and the content of the manuscript is consistent with this journal, and the content is also slightly innovative. may be considered for publication in this journal. Of course, these include but are not limited to errors that need to be revised first.
1. line 532, Doesn't seem like the correct DOI Number?
2. Figure 2/3/5 should be clearer pictures.
3. The size and color of the circles in Figure 4 should have rulers.
Author Response
Reviewer: The authors are trying to develop an R-based method (MetaboAnalystR) for metabolome data processing. Compared with the existing related software, the software shows advantages in the data processing in the manuscript. The manuscript describes a possible open source metabolomics data analysis method, and the content of the manuscript is consistent with this journal, and the content is also slightly innovative. may be considered for publication in this journal. Of course, these include but are not limited to errors that need to be revised first.
Point 1: line 532, Doesn't seem like the correct DOI Number?.
Response 1: Thank you for pointing this out. Corrected.
Point 2: Figure 2/3/5 should be clearer pictures.
Response 2: We redrawed Figures 2/3/5 to ensure higher quality.
Point 3: The size and color of the circles in Figure 4 should have rulers.
Response 3: We added size and color rulers on Figure 4.
Authors: We highly appreciate your precious time in reviewing our paper. Thank you for your feedback and valuable remarks.
Round 2
Reviewer 2 Report
My former comments have been addressed.
Author Response
Authors’ Reply to the Review Report Reviewer: My former comments have been addressed. Authors: The authors are grateful for the time and effort that you dedicated to our manuscript. We have submitted our article for MDPI English language editing to address the remark about English language and style. Thank you for careful reading of our manuscript and constructive feedback.